# Natural 2-Amino-3-Methylhexanoic Acid as Plant Elicitor Inducing Resistance against Temperature Stress and Pathogen Attack

**DOI:** 10.3390/ijms23105715

**Published:** 2022-05-20

**Authors:** He Wang, Jingjing Li, Qian Yang, Lan Wang, Jing Wang, Yaxin Zhang, Yanjing Guo, Rui Li, Ruiqi Zhang, Xiaorong Tao, Bernal E. Valverde, Sheng Qiang, Hazem M. Kalaji, Shiguo Chen

**Affiliations:** 1Weed Research Laboratory, College of Life Science, Nanjing Agricultural University, Nanjing 210095, China; 2016816127@njau.edu.cn (H.W.); 2020216001@stu.njau.edu.cn (J.L.); 2020116004@stu.njau.edu.cn (Q.Y.); 2019816109@njau.edu.cn (L.W.); 2020816107@stu.njau.edu.cn (J.W.); 2019116003@njau.edu.cn (Y.Z.); 2018116080@njau.edu.cn (Y.G.); lirui01@163.com (R.L.); ideatrop@ice.co.cr (B.E.V.); wrl@njau.edu.cn (S.Q.); 2College of Agronomy, JCIC-MCP, National Key Laboratory of Crop Genetics and Germplasm Enhancement, Nanjing Agricultural University, Nanjing 210095, China; zrq@njau.edu.cn; 3Department of Plant Pathology, Nanjing Agricultural University, Nanjing 210095, China; taoxiaorong@njau.edu.cn; 4Department of Plant Physiology, Institute of Biology, Warsaw University of Life Sciences SGGW, Nowoursynowska 159, 02-776 Warsaw, Poland; hazem@kalaji.pl

**Keywords:** agricultural production, natural product, plant resistance inducer, biopesticide, biotic and abiotic stress

## Abstract

2-Amino-3-methylhexanoic acid (AMHA) was synthetized as a non-natural amino acid more than 70 years ago; however, its possible function as an inducer of plant resistance has not been reported. Plant resistance inducers, also known as plant elicitors, are becoming a novel and important development direction in crop protection and pest management. We found that free AMHA accumulated in the mycelia but not in fermentation broths of four fungal species, *Magnaporthe oryzae* and three *Alternaria* spp. We unequivocally confirmed that AMHA is a naturally occurring endogenous (2*S*, 3*S*)-*α*-amino acid, based on isolation, purification and structural analyses. Further experiments demonstrated that AMHA has potent activity-enhancing resistance against extreme temperature stresses in several plant species. It is also highly active against fungal, bacterial and viral diseases by inducing plant resistance. AMHA pretreatment strongly protected wheat against powdery mildew, *Arabidopsis* against *Pseudomonas syringae* DC3000 and tobacco against *Tomato spotted wilt virus*. AMHA exhibits a great potential to become a unique natural elicitor protecting plants against biotic and abiotic stresses.

## 1. Introduction

Plants are constantly subjected to different abiotic and biotic stress conditions throughout their life cycle. For crops, the most characterized abiotic stresses include extreme temperature, drought, salinity and heavy metal toxicity [1]. Biotic stresses are imposed by bacteria, fungi, viruses, insects, parasites and weeds [2]. Severe and sustained stresses can lead to significant decreases in crop yield and even to plant death. Chemical pesticides have made a great contribution to overcoming these biotic stresses and ensuring food safety. According to the Food and Agriculture Organization of the United Nations (FAO), the use of pesticides was increased from 2 million tons in 1990 to 4 million tons in 2019, mainly in agricultural production worldwide (https://www.fao.org/faostat/en/#data/RP/visualize 15 May 2022). However, pesticide resistance, environmental pollution and food safety issues caused by the overuse or inappropriate use of chemical pesticides currently court worldwide attention [3]. Plant elicitors, also called resistance inducers or defense activators, are regarded as an innovative tool to reduce the use of chemical pesticides, offering a new direction for sustainable plant protection [4,5].

Exogenous application of plant elicitors aims at activating the plant defense system to quickly and effectively respond to biotic or abiotic stresses [4,6]. In relation to disease, and different from conventional fungicides, plant elicitors usually do not target the pathogenic agents directly, but at very low concentrations promote plant growth, improve nutrient use efficiency and strengthen the plant’s own defense systems, resulting in a broad-spectrum and long-term increased tolerance to infection [5,7,8]. Thus, plant elicitors are a new concept in prevention of pest attack and pathogen infection as well as damage from abiotic stresses. Plant elicitors include a variety of chemical or biological active ingredients. Biologically active elicitors are mostly derived from seaweed extracts, humic substances and beneficial bacteria or fungi. These active biochemicals usually include proteins, protein hydrolysates, polypeptides, amino acids, glycoproteins, glycopeptides, oligosaccharides, chitosan and lipids [5,9,10,11]. Seaweed extracts rich in amino acids and alginate improved plant tolerance to drought, salinity and heat [12,13]. Alginate oligosaccharide, a hydrolysis product of a natural polysaccharide isolated from brown sea algae, increased fruit quality and induced plant innate immune responses against diseases [14,15]. Chitosan stimulated tomato plant defense system via the nitric oxide (NO) signaling pathway [16].

The *α*-amino acid, 2-amino-3-methylhexanoic acid (AMHA), also known as *β*-methylnorleucine, was chemically synthesized 74 years ago [17]. Later, it was reported that AMHA could be produced by *Serratia marcescens* strain Ar130-15, an *α*-aminobutyric acid-resistant mutant [18] grown in a norvaline-containing medium [19]. The wild-type strain of *S. marcescens*, however, could not produce AMHA in conventional medium without exogenous norvaline and leucine [20]. Muramatsu et al. (2002) found that the genetically engineered *Escherichia coli* strain JM109 could also produce AMHA in culture medium without added amino acids [21]. The synthesis of AMHA in recombinant *E. coli* under controlled conditions proceeded from α-ketovalerate with the enzymes of the isoleucine/valine biosynthetic pathway [22]. Nevertheless, AMHA is still considered as a non-naturally occurring nonproteinogenic amino acid because it has never been detected in normal organisms except mutant or engineering bacteria. A few studies indicate that AMHA at 1 mM concentration has marked antimicrobial activity against *Bacillus subtilis* and *E. coli* K-12, and slightly growth-inhibitory activity in relation to *Achromobacter butyri*, *Arthrobacter ureafaciens*, *E. coli* B and *Pseudomonas aeruginosa* [19,23]. To our knowledge, besides the antagonistic activity to bacteria, there are no other reports about novel bioactivities of AMHA, especially in relation to induction of plant resistance.

In this study, we aimed to confirm AMHA as a naturally occurring amino acid and test its activity-inducing plant resistance. It was found that the filamentous fungi *Alternaria* sp. and *Magnaporthe oryzae* can produce AMHA. After isolation, purification and structural identification, we were able to prove that AMHA is indeed a free natural product in some fungi. Further studies were carried out to assess the activity of AMHA as a plant elicitor. It was suggested that AMHA could significantly induce resistance in plants against high/low temperature stresses as well as against fungal, bacterial and viral infections at very low doses. Thus, AMHA has great potential to be developed as a commercial natural plant resistance inducer.

## 2. Results and Discussion

### 2.1. AMHA Is a Naturally Occurring α-Amino Acid

Although AMHA was successfully isolated in the 1980s from a mutant strain of *S. marcescens* cultivated in medium supplemented with norvaline [19,20], it has long been regarded as an unnatural amino acid. To probe whether fungal strains naturally produce AMHA, extracts from mycelia of *A. alternata* were separated using medium-pressure liquid chromatography (MPLC). The chromatogram of the AMHA standard had a clear peak at a retention time of 4.44 min (Figure 1a). Based on the chromatographic profile for mycelia extracts (Figure 1b), a peak observed at 4.43 min corresponded to that of the AMHA standard. Therefore, *A. alternata* has the ability to produce free AMHA in the absence of norvaline. To further confirm this finding, the fraction of the peak at 4.43 min was collected, purified and crystallized. After white crystals were obtained (Figure 1c), its melting point was determined to be 268 °C (decompose), which was the same in prior research [19]. Subsequently, its chemical structure was identified by the infrared (IR), mass (MS) and nuclear magnetic resonance (NMR) spectra.

The IR spectrum of the fraction was determined for the identification of functional groups. Two peaks observed at about 3400 cm^−1^ and 3325 cm^−1^ were attributed to the asymmetric and symmetric stretching vibration of NH_2_, respectively (Figure 2a). An absorption peak at 2899 cm^−1^ indicated that the fraction had the symmetric stretching vibration of an OH group. The characteristic peak at 1727 cm^−1^ corresponded to the stretching vibration of a C=O group. An additional peak at 1214 cm^−1^ was associated with the stretching vibration of a C-N group. Thus, the crystalized compound had an amino acid structure.

The electrospray ionization mass spectrum of the compound depicted a significant ion peak at a retention time of 1.66 min in the positive mode [M + H]^+^ (Figure 2b). Subsequently, the dominant peak was analyzed with MS spectrum, indicating that the *m*/*z* value of the compound was 146.1172 (Figure 2c). The molecular formula of the compound was deduced as C_7_H_15_NO_2_ (Calculated *m*/*z*: 146.1180) by elemental composition analysis. This was in agreement with the literature [19,21] and data for the AMHA chemical standard.

To further confirm the chemical structure, ^1^H and ^13^C NMR spectra of the compound were detected on the basis of the structure information of standard AMHA. The ^1^H NMR data displayed typical proton signals that included a proton of hydroxyl (OH) at δ_H_ 13.74 (1 H, br), two protons of amino (NH_2_) at δ_H_ 8.39 (2 H, br), two protons of methines (CH) at δ_H_ 3.77 (1H, s, CHNH_2_) and δ_H_ 2.05 (1H, s, CHCH_3_), four protons of methylenes (CH_2_) at δ_H_ 1.51–1.40 (2 H, m, CH_3_CH_2_CH_2_) and δ_H_ 1.26–1.11 (2 H, m, CH_3_CH_2_CH_2_), six protons of methyls (CH_3_) at δ_H_ 0.93 (3 H, d, CH_3_CH) and δ_H_ 0.87 (3 H, t, CH_3_CH_2_) (Figure 2d). A signal at δ_H_ 2.50 was attributed to the protons of the dimethyl sulfoxide (DMSO) solvent. The ^1^H NMR results are in good accordance with an early characterization of AMHA [19]. The obtained ^13^C NMR spectra of the compound exhibited seven absorption peaks attributable to a C=O carbon atom at δ_C_ 170.87 (C-1), two CH carbon atoms at δ_C_ 56.72 (C-2) and δ_C_ 34.95 (C-3), two CH_2_ carbon atoms at δ_C_ 33.87 (C-4) and δ_C_ 20.12 (C-5) and two CH_3_ carbon atoms at δ_C_ 15.38 (C-7) and δ_C_ 14.30 (C-6). Two CH_3_ carbon atoms of solvent DMSO revealed a strong signal at δ_C_ 39.51 (Figure 2e). Together, the NMR data and the IR and MS spectra indicate that the purified compound was AMHA. This unequivocally confirmed that the fungal strain of *A. alternata* growing in conventional medium naturally produced free AMHA.

To confirm the conformation of AMHA, the optical rotation was firstly measured, which was ${\left[\alpha \right]}_{D}^{25}$ + 29.0 (c 0.5, 6 M HCl). Computational calculation of electrostatic circular dichroism (ECD) spectrum of the (2S, 3S)-AMHA was performed using time-dependent density functional theory (TDDFT). Comparison of the computational spectrum with experimental ECD spectrum of AMHA allowed us to determine its absolute configuration. The overall line shape and band positions of the experimental ECD spectrum were in exact agreement with the computational ECD spectrum (Figure 2f). Therefore, it is demonstrated that the absolute configuration of AMHA isolated from *A. alternata* is assigned to (2S, 3S)-2-amino-3-methylhexanoic acid. This result is compatible with a previous report [19].

### 2.2. Free AMHA Is Produced by Different Fungal Species

To further corroborate that AMHA is a naturally occurring product, mycelial extracts of four different fungi cultivated for 1 to 7 days on Czapek medium were used to monitor the dynamics of AMHA production by ultra-performance liquid chromatography mass coupled to a mass spectrometer (UPLC–MS). As before, a strong signal peak of AMHA standard with an [M + H]^+^ ion at *m*/*z* 146.1180 was observed at the 1.66 min retention time (Figure 3a). A calibration curve of the peak area versus standard concentration was plotted to estimate the content of AMHA in mycelia (Figure 3b). All mycelial extracts had peaks corresponding to AMHA (*m*/*z* 146.1180) with retention times of 1.61 min for *A. alternata* and *M. oryzae,* 1.65 min for *A. alternata* f. sp. *lycopersici* and 1.58 min for *A. brassicicola* (Figure 3c). The peak signal intensity of the extracts of *A. alternata* and *M. oryzae* was distinctly stronger than that of another two necrotrophic fungi, *A. alternata* f. sp. *lycopersici* and *A. brassicicola* (Figure 3c). Production of AMHA by mycelia was detected on the second day with a sharp increase up until the fifth day, beginning to plateau at the sixth day. The highest production of free AMHA by *A. alternata* mycelia was reached at 6.41 μg g^−1^ FW on the seventh day (Figure 3d). The amount of AMHA produced by the other species was substantially lower after seven days of culture. *M. oryzae*, *A. alternata* f. sp. *lycopersici* and *A. brassicicola* produced 1.72, 0.94 and 0.51 μg AMHA g^−1^ FW, respectively (Figure 3d). Production of free AMHA by fungal mycelia might be ubiquitous.

Interestingly, typical signals of AMHA with an [M + H]^+^ ion at *m*/*z* 146.1180 were observed at retention times of 1.70 to 1.74 min for the mycelial extracts when the four fungi cultivated for 7 days in Czapek liquid medium (Figure 4a). Conversely, a distinct signal peak of a fragment with *m*/*z* 146.1180 could not be detected in the fermentation broths of these four fungi (Figure 4b). Mycelia of *A. alternata* grown in liquid medium produced substantially more AMHA than the other three fungi. At 7 days of culture, the content of AMHA in the mycelia of *A. alternata*, *M. oryzae*, *A. alternata* f. sp. *lycopersici* and *A. brassicicola* was 4.04, 1.12, 0.66 and 0.34 μg g^−1^ FW, respectively (Figure 4c). It is likely that AMHA is an endogenous secondary metabolite naturally produced by fungal mycelia because free AMHA was not detected in their fermentation broths (Figure 4c). Comparatively, AMHA was isolated from the fermentation broths of *S. marcescens* mutant strain Ar130-1 as an exocrine secretion product in the presence of norvaline [20]. After 72 h incubation, the yield of AMHA in the broths was over 2 mg mL^−1^, which is much higher than that obtained with the fungi in our study. In addition, AMHA was not detected in the leaves of *Arabidopsis*, winter wheat or rice (Appendix A).

### 2.3. AMHA Induces Plant Resistance to Temperature Stress

Amino acids are known for having positive effects on plant growth, development and yield because of their role in biosynthesis of proteins, pigments, vitamins, coenzymes, purine and pyrimidine bases and other non-protein nitrogenous compounds [24]. For example, γ-aminobutyric acid, a non-protein amino acid, is involved in regulation of pollen tube growth to the ovary, C:N balance, cytosolic pH, oxidative stress and stress response, acting as a signaling molecule in plants [25]. γ-Aminobutyric acid accumulates rapidly in tissues when plants are exposed to both biotic (insect herbivory, pathogen infection) or abiotic (heavy metal, salinity, heat, cold, drought, anoxia, wounding) stresses, showing specific defensive roles [25,26]. Besides γ-aminobutyric acid, there are other amino acids, such as β-aminobutyric acid, glutamate and proline, that have also been proposed to play relevant roles in plant defense [27,28]. As a natural α-amino acid, whether AMHA play any important role in plant response to stress is completely unknown.

Temperature is a key factor governing plant growth and development. Extreme temperature is a frequent stress in agriculture, severely delaying crop growth and development, reducing production and quality and even causing plant death [29]. To investigate the possible role of AMHA as an inducer of plant resistance against temperature stress, seedlings of bentgrass, wheat, tomato and strawberry were treated with this amino acid before being subjected to extreme temperatures. Temperature stress in these plants is one of the most common and major production problems.

The high temperature regime severely affected, almost to death, the three crop plants, bentgrass, wheat and tomato (mock treatment with only of 0.02% Tween-20), compared with the control not exposed to heat stress (Figure 5a). AMHA pretreatment significantly increased the heat resistance of these three plants relative to the mock (Figure 5a). The heat injury index of wheat and tomato seedlings treated with increased concentrations of AMHA dramatically decreased under high temperature conditions compared with the mock (Figure 5b). At 10,000 nM AMHA, most wheat and tomato plants remained alive (Figure 5a) with 60% and 40% lower heat injury index than the mock, respectively (Figure 5b). For bentgrass, high temperature caused severe chlorosis, shriveling and death of seedlings when treated only with 0.02% Tween-20 (Figure 5a). AMHA enhanced plant resistance against heat damage even at 10 nM (Figure 5a,b). Furthermore, shoot growth of the three crop plants was totally inhibited by high temperature conditions. Exogenous pretreatment with 10,000 nM AMHA distinctly alleviated such growth inhibition, resulting in 88%, 78% and 438% increases in shoot fresh weight for bentgrass, wheat and tomato, respectively, compared to the corresponding mock after 7 d of recovery from heat treatment (Figure 5a,c). Based on the severity of heat damage and total aboveground biomass, the best AMHA concentration to alleviate heat damage and enhance heat resistance was 100 nM for bentgrass, 10,000 nM for wheat, 1000 to 10,000 nM for tomato, respectively. Evidently, AMHA has high capability of inducing plant resistance against heat stress.

Tomato plants treated with 0.02% Tween-20 and subjected to low temperature conditions (−2 °C) suffered freezing damage on some leaf margins and tips, developing slightly rolling and wilting of leaves at 5 d of recovery after cold stress (Figure 6a). Strawberry plants exposed for 12 h at −4 °C and recovered for 5 d showed their leaves to be severely wilted or necrotic (Figure 6a). AMHA pretreatment visibly reduced damage from cold stress in both tomato and strawberry plants (Figure 6a,b). Compared with the mock, the cold injury index was reduced by 34%, 66% and 66% in tomato, and 50%, 70% and 80% in strawberry by the pretreatment with 10, 100 and 1000 nM AMHA, respectively (Figure 6b). Moreover, the pretreatment with AMHA significantly alleviated shoot growth inhibition caused by cold stress (Figure 6a,c). At 1000 nM AMHA pretreatment, the aboveground biomass of tomato and strawberry increased by 24% and 84%, respectively, compared to the mock, which reached over 70% of the control plants not exposed to cold stress (Figure 6c). This suggests that AMHA is also active against cold stress in several low-temperature sensitive plants.

### 2.4. AMHA Induces Plant Resistance to Pathogen Infection

Powdery mildew, caused by the fungal pathogen *B. graminis*, is one of the most destructive wheat diseases worldwide [30]. At 8 dpi, abundant grayish white patches of *B. graminis* had grown on the leaves of susceptible wheat plants pretreated with 0.02% Tween-20 (mock), but much less fungal growth was observed on those wheat leaves pretreated with AMHA at increasing concentrations (Figure 7a,b). Wheat plants were much stronger and straighter in the presence of AMHA compared with the mock as a result of decreased fungal growth on the surfaces of leaves and stems (Figure 7a,b). This suggests that AMHA pretreatment conferred wheat plants resistance to powdery mildew. Consistent with this observation, extensive chlorotic lesions were formed on the leaves of mock-treated wheat plants, and only a few chlorotic spots were found on the leaves of AMHA-treated plants (Figure 7c). These results were further supported by data of chlorophyll fluorescence imaging (Figure 7d). Images of maximum PSII quantum yield (F_V_/F_M_) can be used as a sensitivity indicator to visualize early tissue damage during pathogen infection [31]. *B. graminis* resulted in fading images of F_V_/F_M_ in mocked-treated wheat leaves, indicating serious foliar damage caused by the pathogen. Pretreatment with AMHA prevented such damage in a concentration-dependent manner. At the highest concentration of 100 μM AMHA, only slight damage was found on the wheat leaves (Figure 7d). This was in good agreement with the status of *B. graminis* grown on wheat leaves (Figure 7b). Our early experiments demonstrated that AMHA could not inhibit the growth of fungal pathogens on PDA medium, even at 1 mM concentration, including *M. oryzae*, *Rhizoctonia solani*, *Fusarium moniliforme*, *F. graminearum*, *R. cerealis*, *Botrytis cinerea* and *Sclerotinia sclerotiorum* (Appendix A). This suggested that AMHA has no antifungal activity. Thus, we conclude that AMHA can effectively elicit wheat resistance against powdery mildew by suppressing the fungal growth and infection on plants.

To test the activity of AMHA against bacterial disease, the model pathogen *P. syringae* DC3000 was used to infect *Arabidopsis* plants. *P. syringae* is the one of the most common plant pathogens because it infects almost all economically important crops [32,33]. *P. syringae* DC3000 also causes bacterial speck disease in its natural host, tomato [34]. *P. syringae* DC3000 caused severe chlorotic lesions in most leaves of mock-treated *Arabidopsis* plants at 7 dpi (Figure 7e). AMHA pretreatment greatly reduced chlorosis severity by inducing strong resistance in *Arabidopsis* against this bacterial disease. The level of plant resistance against *P. syringae* DC3000 gradually increased with AMHA concentration (Figure 7e). To estimate the effect of AMHA on bacterial growth, the number of bacteria in *Arabidopsis* plants was also measured after 4 days of *P. syringae* DC3000 inoculation. A concentration-dependent decrease in the bacterial population was observed in the presence of AMHA. Total bacterial number concentration of fresh *Arabidopsis* plants pretreated with 10, 100, 1000 and 10,000 nM AMHA, respectively, reduced by 30%, 51%, 65% and 95% compared to the mock-pretreated plants (Figure 7f). Previous studies showed that 1 mM concentration of AMHA were only slightly antagonistic to *P. aeruginosa* and were not obviously antagonistic to *P. fluorescens* [19]. Clearly, bacterial growth was strongly suppressed in AMHA-pretreated plants, demonstrating that AMHA-induced resistance response of *Arabidopsis* plants triggered protection against *P. syringae* DC3000. These results confirm that AMHA can activate plant immunity against the virulent *P. syringae* DC3000 due to the inhibition of bacterial growth in plant tissues.

Besides fungi and bacteria, virus are also important plant pathogens. On the basis of scientific and economic importance, Tomato spotted wilt virus (TSWV) is among one of most destructive plant viruses in the world [35]. TSWV has global distribution and a very wide host range of over 1000 plant species from more than 80 families, causing crop losses at above 1$ billion annually [35,36]. To investigate whether AMHA can protect plants against TSWV infection, tobacco leaves were sprayed with AMHA followed with the infiltration of the Agrobacterium carrying TSWV SR_(+)eGFP_ + M_(−)opt_ + L_(+)opt_ constructs. At 7 days post TSWV inoculation, eGFP fluorescence expression of mock-pretreated tobacco plants was highly abundant in a group of cells; however, eGFP fluorescence of AMHA-pretreated plants was sporadically expressed in leaf tissues (Figure 7g). The average intensity of eGFP fluorescence signal was significantly decreased by 37%, 45% and 46% in leaves pretreated with 0.1, 1 and 10 nM AMHA compared to the mock-treated leaves, respectively (Figure 7h). These results suggest that AMHA pretreatment could also effectively induce plant defense against TSWV infection even at a very low concentration.

## 3. Materials and Methods

### 3.1. Fungus Strains and Chemicals

The *A. alternata* (Fr.) Keissler wild-type strain NEW001 was isolated from the invasive plant *Ageratina adenophora* [37]. *A. alternata* f. sp. *lycopersici* and *A**. brassicicola* were obtained from the China General Microbiological Culture Collection Center (Beijing, China). The strain of *M. oryzae* was provided by M. Zhou (College of Plant Protection, Nanjing Agricultural University (NAU), China). *P. syringae* pv. *tomato* strain DC3000 (*P. syringae* DC3000) was a gift from Z. Wei (College of Resources and Environmental Science, NAU, Nanjing, China).

DMSO-*d*_6_ and tetramethylsilane (Me_4_Si, TMS) were purchased from Aladdin Biochemical Technology Co., Ltd. (Shanghai, China). The standard AMHA (>98% purity) was obtained from Shanghai Shunxin Science and Technology Ltd. (Shanghai, China). Acetonitrile, agar, ammonium acetate (NH_4_Ac) and formic acid were purchased from Sigma-Aldrich (Shanghai, China). Yeast extract was from Oxoid Ltd. (Basingstoke, Hants, UK). Other chemicals were purchased from Sinopharm Chemical Reagent Co., Ltd. (Shanghai, China).

The stock solution of AMHA isolated from *A. alternata* at 1 mM was prepared with distilled water.

### 3.2. Cultivation of Fungi

Fungal strains were maintained on potato dextrose agar (PDA) medium. Cultivation experiments were performed on Czapek medium as previously described [38]. The Czapek medium (pH 5.5) contains 40 g D-glucose, 1 g NaNO_3_, 0.25 g NH_4_Cl, 1 g KH_2_PO_4_, 0.25 g KCl, 0.25 g NaCl, 0.5 g MgSO_4_·7H_2_O, 0.01 g FeSO_4_·7H_2_O, 0.01 g ZnSO_4_·7H_2_O, 1 g yeast extract, 15 g agar (for solid medium) and enough distilled water to make one liter. A 5-mm-diameter agar disc was taken from the margin of mycelia of *A. alternata*, *A. alternata* f. sp. *lycopersici*, *A. brassicicola* and *M. oryzae* grown on PDA medium, and transferred to a new 100-mm-diameter Petri dish (Corning, NY, USA) containing 20 mL of Czapek solid medium. Fungi were allowed to grow at 25 °C for 7 days in darkness. For liquid cultivation, five 5-mm-diameter agar discs taken from the margin of four strains grown on PDA medium were transferred to a new 250 mL flask containing 100 mL of Czapek liquid medium and grown at 25 °C, 140 rpm for 7 days in darkness. Fresh mycelia and Czapek fermentation broths were separated for further experiments by centrifugation at 1000× *g* for 10 min (Allegra^TM^ 64R centrifuge, Beckman, Coulter, CA, USA).

### 3.3. Extraction, Purification and Structure Analysis

After 7 days of growth in Czapek solid medium, 1 g *A. alternata* mycelia were harvested and homogenized in liquid nitrogen. The mycelial powder was put into a 10 mL tube containing 5 mL distilled water and completely mixed by shaking for 60 min to extract AMHA at room temperature.

The supernatant was transferred to a new 10 mL tube after centrifugation at 5500× *g* for 10 min. The crude extracts were dried for 24 h under a vacuum in an LGJ-10 lyophilizer (Xinyi Inc., Beijing, China) and redissolved in 1 mL distilled water. The extract was filtered through a GF/B 0.22-μm glass microfiber filter (Whatman, Kent, UK) and stocked in a clean 2 mL tube. The isolation and purification of AMHA was performed on a MPLC (NGC^TM^ Quest 10 Plus Chromatography System, Bio-Rad Laboratories, Inc., Hercules, CA, USA) equipped with a SunFire^TM^ Prep C_18_ column (10 × 250 mm, 10 µm) (Waters, MA, USA). The detection wavelength was set at 210 nm. The mobile phase was a water solution containing 0.1% formic acid (*v*/*v*) and acetonitrile (60: 40, *v*/*v*). For each separation, 50 μL samples were injected and passed at a flow rate of 2.0 mL min^−1^ at ambient temperature. The major components with a retention time of 4.4 min as the standard were collected and freeze-dried for 24 h under vacuum to obtain the purified white crystals. The melting point of white crystals was measured with a MPA100 automated melting point system (Stanford Research Systems, California, CA, USA). Optical rotation was determined using an Autopol IV automatic polarimeter (Rudolph Research Analytical, Newburgh Road Hackettstown, Hackettstown, NJ, USA) with 6 M HCl as solvent. ECD spectra were measured by a JASCO J-1500 CD Spectrophotometer (JASCO Co., Tokyo, Japan).

The IR spectra of the crystals were recorded according to Bagheri et al. (2018) [39]. The IR data were acquired on an FTIR system (Nicolet^TM^ 460 FT-IR spectrophotometer, Thermo Fisher Scientific, Waltham, MA, USA). IR data were analyzed on OMNIC software (Thermo Fisher Scientific).

The crystals were analyzed using an acquity UPLC system (Waters, Milford, MA, USA) coupled to a Waters G2-XS Q-TOF mass spectrometer, as previously described with minor modifications [40]. UPLC separations were performed with a binary solvent delivery system and an auto-sampler on a Waters Acquity UPLC BEH C_18_ column (2.1 mm × 100 mm, 1.7 μm particles). The flow rate was 0.4 mL min^−1^. The injection volume of samples was 2 μL. The detection wavelength was 280 nm. Two consecutive isocratic mobile phase mixtures consisted of water containing 0.1% formic acid and 2 mM NH_4_Ac (solvent A) and acetonitrile containing 0.1% formic acid and 2 mM NH_4_Ac (solvent B). The gradient program was as follows: within 14 min, 5% solvent B was held for 1 min, then 5% solvent B was increased within 9 min to 95% and held for an additional 2 min and then 95% solvent B was decreased within 1 min to 5%, which was held for 1 min.

Mass detection was performed using electrospray source in positive ion mode with MS acquisition mode at a selected mass range of 50–1200 *m*/*z*. The lock mass option was enabled using leucine-enkephalin (*m*/*z* 556.2771) for recalibration. The ionization parameters were set as follows: capillary voltage of 3.0 kV, cone voltage of 30 V, source temperature at 120 °C, and desolvation gas temperature at 400 °C. The collision energy was 20–40 eV. Data acquisition and processing were performed using Masslynx 4.1 (Waters, MA, USA). Extraction of centroid spectra peaks with a width of 0.01 Da was used to determine the extracted ion chromatograms (EICs) from the total ion chromatogram (TIC).

Nuclear magnetic resonance (NMR) spectra were recorded on a Bruker AM-400 spectrometer (Bruker, Zuerich, Switzerland). Chemical shifts were given in parts per million (*δ*) with reference to the residual DMSO-*d*_6_ (^1^H, *δ* = 2.50 ppm, ^13^C, *δ* = 39.51 ppm) signal. Me_4_Si was used for the internal standard. Coupling constants (*J*) are given in Hz. Fifteen mg purified AMHA was dissolved in 0.5 mL DMSO-*d*_6_ for NMR. NMR data were analyzed MestReNova 11.0 software (Mestrelab Research SL, Santiago de Compostela, Spain).

### 3.4. The Physical and Spectroscopic Data of AMHA

AMHA: The compound was white crystals; mp 268 °C (decompose); ${\left[\alpha \right]}_{D}^{25}$ +29.0 (*c* 0.5, 6 M HCl); ECD (1.0 × 10^−4^ M, H_2_O) Δ*ε*_201_ = +31.71; IR (KBr) ν_max_ 3400, 3325, 2899, 1727, 1214 cm^–1^; ^1^H NMR (DMSO-*d*_6_, 400 MHz): *δ* 13.74 (1H, br, O*H*), 8.39 (2H, br, N*H*_2_), 3.77 (1H, s, C*H*NH_2_), 2.05 (1H, s, C*H*CH_3_), 1.51–1.40 (2H, m, CH_3_CH_2_C*H*_2_), 1.26–1.11 (2H, m, CH_3_C*H*_2_CH_2_), 0.93 (3H, d, *J* = 8.0 Hz, C*H*_3_CH), 0.87 (3H, t, *J* = 8.0 Hz, C*H*_3_CH_2_); ^13^C NMR (DMSO-*d*_6_, 100 MHz): *δ* 170.87 (C, *C=*O), 56.72 (CH, *C*HNH_2_), 34.95 (CH, *C*HCH_3_), 33.87 (CH_2_, CH_3_CH_2_*C*H_2_), 20.12 (CH_2_, CH_3_*C*H_2_CH_2_), 15.38 (CH_3_, *C*H_3_CH), 14.30 (CH_3_, *C*H_3_CH_2_); HR–MS m/z 146.1172 [M + H]^+^ (calcd for C_7_H_15_NO_2_, 146.1181).

### 3.5. Conformational Analysis, Geometrical Optimization and ECD Calculation of (2S, 3S)-AMHA

The absolute configuration analysis of AMHA was carried out on the basis of the systematic conformation search using Molecular Operating Environment (MOE) software version 2016 (Chemical Computing Group Inc., Montreal, QC, Canada) under the MMFF94 force field. An energy cut of 2.5 kcal·mol^−1^ was chosen in order to select a wide distribution of conformers. The stable conformers subjected to ECD calculation were re-optimized using density functional theory (DFT) at the B3LYP/6-31G (d) level in Gaussian 09 software (Gaussian Inc., Wallingford, CT, USA) [41]. Seven lowest conformers were selected for theoretical calculation of ECD, which was performed using TDDFT method at the B3LYP/6-311+G (d, 2p) level with CPCM solvation model in water. The ECD spectra were simulated using the overlapping Gaussian function by applying a half-bandwidth at 1/e peak height, σ = 0.20 for all. The final computational ECD spectra were obtained according to the Boltzmann-calculated contribution of each conformer.

### 3.6. Test of AMHA in Different Fungus Strains and Plants

To quantitatively analyze the yield of AMHA in *A. alternata*, *A*. *alternata* f. sp. *lycopersici*, *A*. *brassicicola* and *M*. *oryzae*, 1 g fresh mycelia of each of the four strains grown in Czapek solid or liquid medium were collected. Extraction of AMHA in mycelia and UPLC–MS analysis were carried out in the same way described before. The content of AMHA in Czapek fermentation broths of four trains was also measured. After centrifugation, the separated fermentation broths of each strain were transferred to a new 250 mL flask. Then, 1 mL of fermentation supernatant was taken and filtered through a 0.22-μm glass microfiber filter (Whatman, Kent, UK). The filtrate was collected in a new 2 mL tube. All the samples were transferred to new 2 mL short thread vials with pre-slit blue short screw (Beijing Labgic Technology Co., Ltd., Beijing, China) before UPLC–MS analysis. Quantification of natural AMHA was performed by employing the external calibration curve. The standard AMHA was dissolved in distilled water to 100 mg L^−1^ as stock solution. The working standard solutions were prepared at 0.25–8 mg L^−1^ from the stock solution by gradient dilution with distilled water.

To conform whether AMHA could be produced in plants, 1 g fresh leaves of 21-day-old *Arabidopsis thaliana* ecotype Col-0 plants, two-leaf stage seedlings of winter wheat (*Triticum aestivum*) and rice (*Oryza sativa*) grown in greenhouse were collected. Extraction in plants and UPLC–MS analysis were carried out in the same way described for fungi.

### 3.7. High/Low-Temperature Resistance Assays

Seeds of winter wheat (*T. aestivum*), bentgrass (*Agrostis capillaris*) and tomato (*Lycopersicon esculentum*) cultivar “Zhongsu 4” were purchased from Jiangsu Mingtian Seed Science and Technology Co. (Nanjing, China). Thirty wheat seeds, 0.2 g of bentgrass seeds and three tomato seeds per pot were planted separately in peat-vermiculite mixtures (1:1, *v*/*v*) in 12-cm-diameter plastic pots and grown at 25 °C under 200 μmol m^−2^ s^−1^ illumination with 12 h photoperiod. Strawberry (*Fragaria ananassa*) cultivar “Hongyan” seedlings at the 5-leaf stage were obtained from T. Gu (College of Horticulture, NAU, China) and then transplanted to 12-cm-diameter pots (one plant per pot) containing peat-vermiculite substrate (1:1, *v*/*v*) and cultivated for 7 days at 25 °C under 200 μmol m^−2^ s^−1^ light for 12 h. Fourteen-day-old wheat seedlings, twenty-five-day-old bentgrass seedlings and eighteen-day-old tomato seedlings were pretreated with AMHA before high temperature treatment. The adapted strawberry seedlings and 45-day-old tomato seedlings were treated with AMHA prior to subjecting them to low temperature stress. Plants were sprayed twice using distilled water plus 0.02% (*v*/*v*) Tween-20 (mock) or AMHA (10, 100, 1000 and 10,000 nM) plus 0.02% (*v*/*v*) Tween-20 with a 24-h interval by a mist sprayer (SKS Bottle & Packaging Inc., Saratoga Springs, NY, USA) until the leaf surfaces were completely wet.

After 24 h of the second spraying, the pretreated bentgrass and wheat plants were exposed to 45 °C for 9 h under 200 µmol m^−2^ s^−1^ illumination and transferred to their normal growth conditions for 7 days for recovery. The pretreated tomato seedlings were subjected to 42 °C for 9 h under light with 200 µmol m^−2^ s^−1^ and recovered for 7 days. For low temperature treatment, the pretreated tomato and strawberry plants, respectively, were subjected to −2 °C for 24 h and −4 °C for 12 h under light of 200 µmol m^−2^ s^−1^ and recovered for 5 days under normal growth conditions. After recovery, plants were photographed using a digital camera (Canon model SX11S, Tokyo, Japan). Heat injury level per plant was rated on a 0–5 scale: 0, no damage; 1, less than 25% of leaf area wilted; 2, 25%–50% of leaf area wilted or yellow; 3, 50%–75% of leaf area wilted or yellow; 4, more than 75% of leaf area wilted or yellow; 5, whole plant died. Cold injury degree per plant was recorded with the following scale: 0, no damage; 1, leaf edge slightly wilted; 2, wilting less than 25% of total leaf area; 3, 25%–50% of leaf area wilted and several spots; 4, more than 50% of leaf area wilted or spots; 5, leaf completely wilted. All leaves of each plant were assessed for injury. The injury index was calculated according to the formula: injury index = 100 × ∑(*v* × *n*)/(5 × *N*), here *n* is the number of plants in a scale, *v* is the scale value, and *N* is the total number of plants tested. Finally, the seedlings of each pot were collected by cutting the stems just above soil surface for fresh weight determination. Plants of four pots were used in each experiment as four independent biological replicates.

### 3.8. Wheat Resistance to Fungus Powdery Mildew

Wheat powdery mildew (*Blumeria graminis* f. sp. *tritici*) (*Bgt*) strains were maintained on seedlings of susceptible wheat variety NAU0686 in a spore-proof glasshouse. Sixty wheat seeds were planted in a soil-peat (3:1, *v*/*v*) substrate mixture in 18-cm-diameter plastic pots at 22 °C under 100 µmol m^−2^ s^−1^ light with 16: 8 h photoperiod and 70% relative humidity in a controlled culture room. Sixty wheat seeds were sown per pot. The seedlings at the two-leaf stage were sprayed three times with AMHA solutions (15 mL per pot) at different concentrations (1, 10 and 100 μM) containing 0.02% Tween-20 or with 0.02% Tween-20 alone (mock) with a 24-h interval using a SKS mist sprayer. After 24 h following the third spraying, the pretreated seedlings were inoculated with powdery mildew conidia (about 15 to 20 conidia on leaf surface per cm^2^) according to Wang et al. (2014) [30], and then incubated at 22 °C under 100 µmol m^−2^ s^−1^ with 16-h photoperiod. Eight days after inoculation, whole plants and disease symptoms of representative leaves were photographed as before. Simultaneously, chlorophyll fluorescence images of F_V_/F_M_ of these representative leaves were also measured under a pulse-modulated Imaging-PAM M-series fluorometer (MAXI-version, Heinz Walz GmbH, Effeltrich, Germany), as previously described [31]. Plants of three pots were used in each experiment as three independent biological replicates.

### 3.9. Arabidopsis Resistance to Bacterial Infection

*A. thaliana* ecotype Col-0 was grown in a peat-vermiculite substrate mixture (1:3, *v*/*v*) at 22 °C under 16 h photoperiod with a light-intensity of 100 μmol m^−2^ s^−1^. Three-week-old healthy plants were sprayed twice to runoff with 0.02% Tween-20 (mock) or four concentrations (10, 100, 1000 and 10,000 nM) of AMHA solution containing 0.02% Tween-20 with a 24-h interval using a SKS mist sprayer, and then incubated for 24 h after a second spraying under the same growth conditions before inoculation with bacteria. *P. syringae* DC3000 was cultured on Luria-Bertani (LB) media with 50 μg mL^−1^ rifampicin at 28 °C for 48 h. Bacteria were collected from LB media and suspended in 10 mM MgCl_2_ solution. Bacterial cell densities (OD_600_) were adjusted to 0.2 (equal to 5 × 10^8^ cells mL^−1^) and then further diluted with 10 mM MgCl_2_ solution with 0.02% Tween-20 to 5 × 10^6^ cells mL^−1^ for inoculation. The pretreated *Arabidopsis* plants were inoculated with *P. syringae* DC3000 suspension and incubated under continuous light (about 100 μmol m^−2^ s^−1^) at 22 °C. Four days post-inoculation (dpi), leaves were collected from at least three different plants (four leaves per plant) and weighed. Subsequently, leaves were homogenized to measure bacterial population, as previously described [34]. The colony forming units (CFU) were normalized as CFU mg^−1^ using the fresh weight of leaves. Seven days after inoculation, disease phenotypes of whole plants were also observed and photographed. Each experiment has three pots with four plants in each.

### 3.10. Tobacco Resistance to Virus Infection

Tobacco (*Nicotiana benthamiana*) plants were grown at 25 °C under 100 µmol m^−2^ s^−1^ light with 16 h photoperiod. Prior to virus infection, 8-week-old plants were pretreated twice by spraying with 0.02% Tween-20 (mock) or 0.1, 1 and 10 nM AMHA solution containing 0.02% Tween-20 with a 24-h interval using a SKS mist sprayer, and then incubated 24 h after second spraying. *Tomato spotted wilt virus* isolate from asparagus lettuce (*Lactuca sativa*) (TSWV-LE) was used in this work [36]. Leaves of the pretreated tobacco plants were infiltrated with a 1:1:1 mixture of *Agrobacterium*, respectively, carrying infectious clones of TSWV SR_(+)eGFP_, M_(−)opt_ and L_(+)opt_. The infectious clones of TSWV were described by Feng et al. (2020) [36]. Seven days post infiltration, the accumulation of TSWV in the infected leaves was visualized by monitoring eGFP fluorescence expression from the SR_(+)eGFP_ replicon under an Axio Imager M2 fluorescence microscope (Carl Zeiss Microscopy GmbH, Oberkochen, Germany). The fluorescence images were quantified using Image J v1.8.0 software (National Institute of Health, Bethesda, Rockville, MD, USA). Plants of three pots were used in each experiment as three independent biological replicates.

### 3.11. Statistical Analysis

One-way ANOVA was carried out and means were separated by Duncan LSD at 95% using SPSS Statistics 26.0 (IBM^®^, Armonk, NY, USA).

## 4. Conclusions

Firstly, it has been demonstrated that AMHA is a naturally occurring endogenous *α*-amino acid since free AMHA can be produced by fungi including *M. oryzae* and three *Alternaria* species in a conventional medium. Furthermore, AMHA indeed exhibits excellent activity against extreme temperature stress as well as against fungal, bacterial and viral diseases by enhancing plant resistance at the concentration range of 0.1–100,000 nM. Undoubtedly, AMHA is a unique plant elicitor with great potential.

Before AMHA can be commercialized as a crop protection product, it is advisable to determine its physio-biochemical and molecular mechanisms for inducing plant resistance against abiotic and biotic stresses. Additionally, greenhouse and field experiments are also essential for assessing the activity spectrum of AMHA as a potent plant elicitor and developing practical technical frameworks for its use in crop protection.

## 5. Patent

The authors have obtained a Chinese patent (ZL202011549486.6) based on the results reported in this paper.

## Figures and Tables

**Figure 1 ijms-23-05715-f001:**
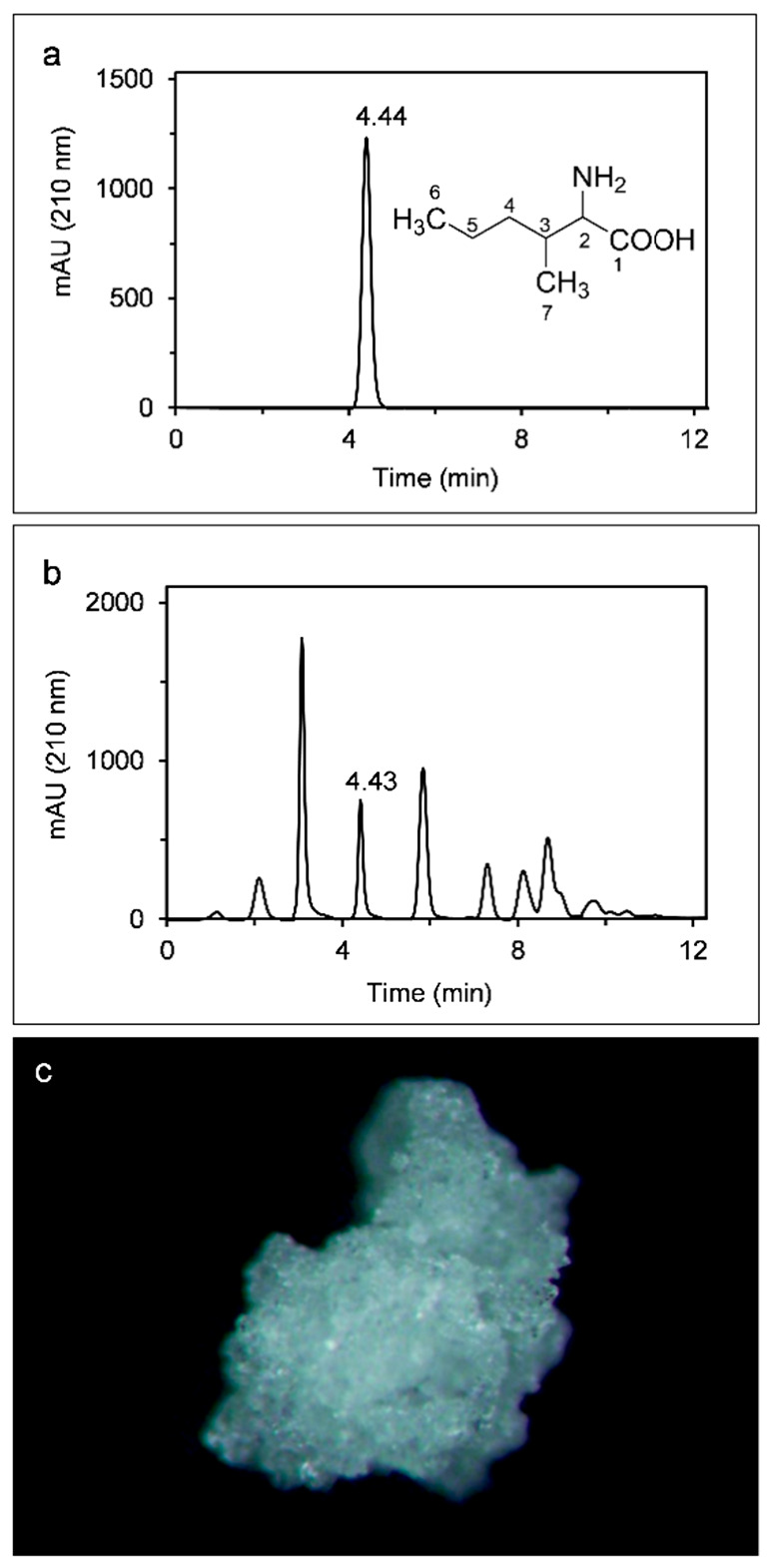
Purification of AMHA isolated from the mycelia of *A. alternata*. (**a**) MPLC chromatogram of 8 mg L^−1^ standard solution of AMHA. (**b**) MPLC chromatogram of extracts from *A. alternata*. The peak corresponding to AMHA is labelled. (**c**) Crystals of naturally produced AMHA.

**Figure 2 ijms-23-05715-f002:**
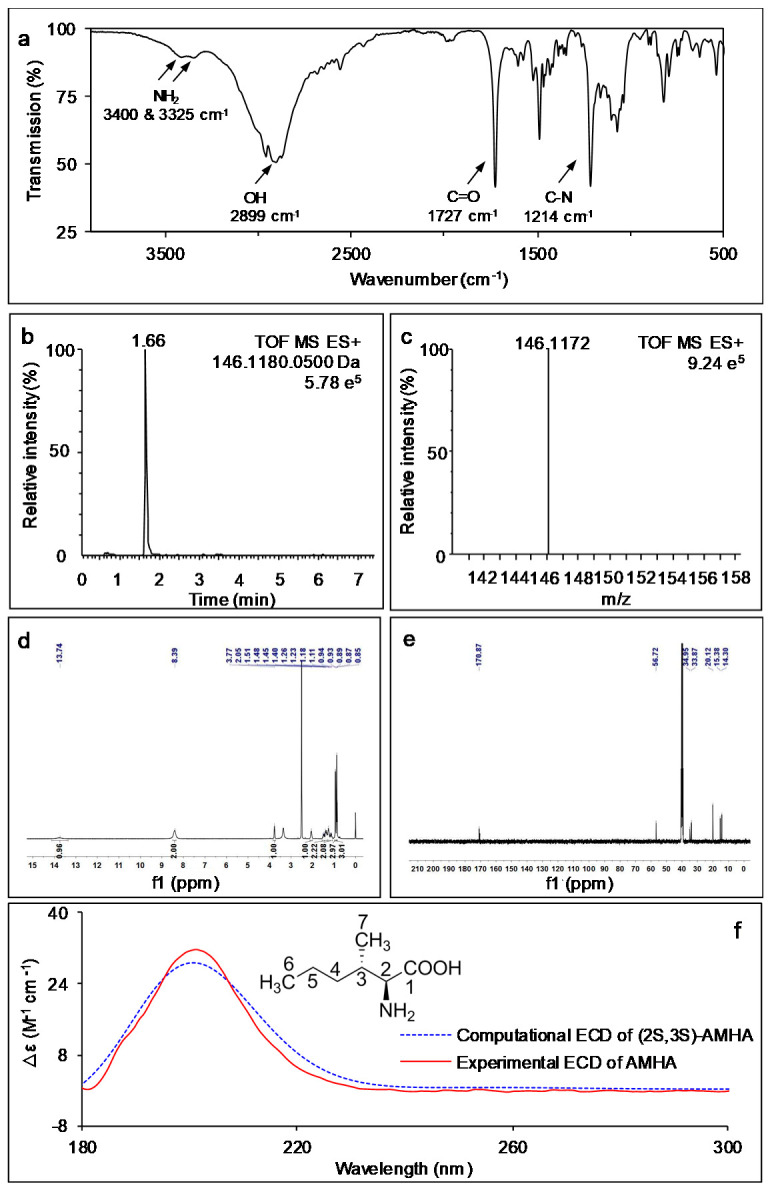
Spectra of naturally produced AMHA isolated from the mycelia of *A. alternata*. (**a**) IR spectrum. (**b**) UPLC–MS chromatogram in positive ion mode. (**c**) Mass spectrum. (**d**) ^1^H NMR spectrum (400 MHz, DMSO-d_6_). (**e**) ^13^C NMR spectrum (100 MHz, DMSO-d_6_). Each spectrum is one of three independent biological replicates with similar results. (**f**) Comparison of experimental and computational ECD spectra of AMHA isolated from *A. alternata*.

**Figure 3 ijms-23-05715-f003:**
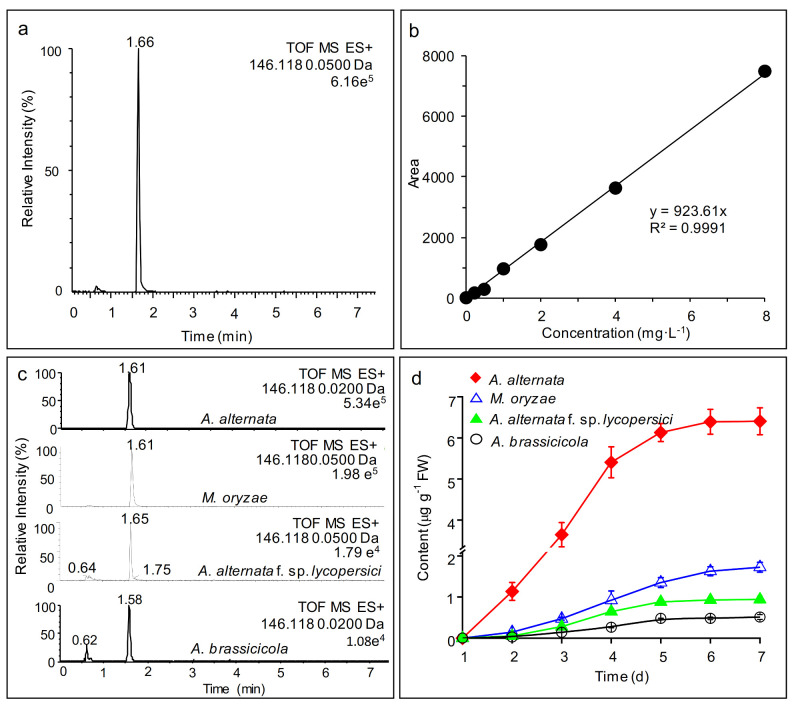
Quantification of AMHA produced by mycelia of four different fungi cultivated on Czapek solid medium. (**a**) UPLC–MS chromatogram of standard AMHA with 8 mg L^−1^ in positive ion mode. (**b**) Calibration curve of the peak signal area at 1.66 min vs. concentration of AMHA standard (mg L^−1^). (**c**) UPLC–MS chromatograms in positive ion mode of the extracts from 7-day-old mycelia of *A. alternata*, *M. oryzae*, *A. alternata* f. sp. *lycopersici* or *A. brassicicola*. (**d**) AMHA production by mycelia of four fungi during the first 7 d of culture on Czapek solid medium. Results represent three independent biological replicates. Data in D represent mean values ± SD.

**Figure 4 ijms-23-05715-f004:**
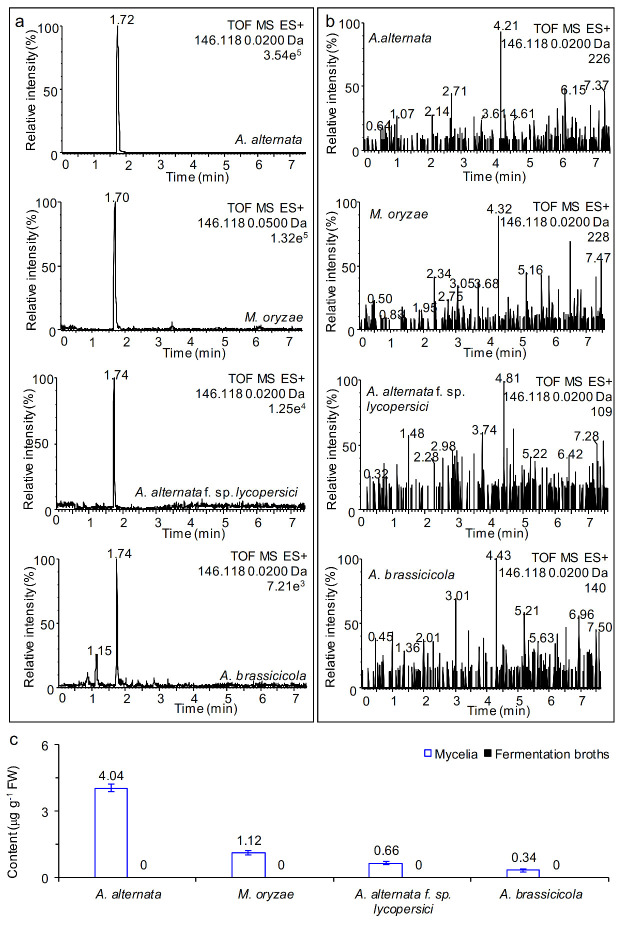
Natural production of AMHA by mycelia and fermentation broths of four fungi. *A. alternata*, *M. oryzae*, *A. alternata* f. sp. *lycopersici* and *A. brassicicola* cultivated for 7 d on Czapek liqiud medium. Results represent three independent biological replicates. (**a**) UPLC–MS chromatogram in positive ion mode of extracts of mycelia of four fungi. (**b**) UPLC–MS chromatogram in positive ion mode of filtrate of fermentation broths of four fungi. (**c**) Content of natural AMHA in the mycelia and fermentation broths of four different fungi at 7 d of cultures on Czapek liquid medium.

**Figure 5 ijms-23-05715-f005:**
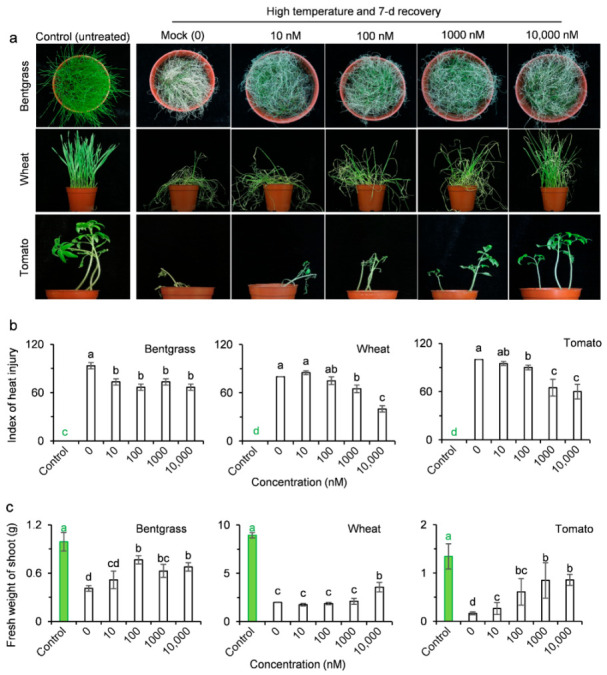
Induction of resistance to high temperature by AMHA in plant seedlings of three crops. After pretreatment with a water dilution of Tween-20 at 0.02% (mock) or different concentrations of AMHA containing 0.02% Tween-20 (10, 100, 1000 and 10,000 nM), seedlings were incubated for 9 h at 45 °C (bentgrass and wheat) or 42 °C (tomato) under 200 µmol m^−2^ s^−1^ light and recovered at normal temperature for 7 d. Seedlings without any treatment, kept in normal growth condition, were used as untreated controls. (**a**) Phenotypes of plants after high temperature treatment. Photographs were taken after recovery for 7 d. (**b**) Effect of AMHA pretreatment on the heat injury index of three plant species treated with high temperature. (**c**) Effect of AMHA pretreatment on the shoot fresh weight of three plant species treated with high temperature. Values are the means ± SD of four independent biological replicates. Different small letters above error bars indicate significance at 0.05 level.

**Figure 6 ijms-23-05715-f006:**
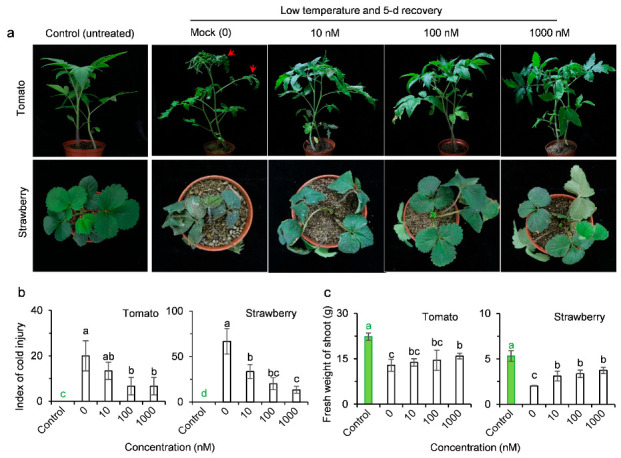
Induction of resistance to low temperature by AMHA in strawberry and tomato seedlings. After pretreatment with water containing 0.02% Tween-20 (mock) or different concentrations of AMHA (10, 100 and 1000 nM) supplemented with 0.02% Tween-20, tomato and strawberry seedlings were treated for 24 h at −2 °C and for 12 h at −4 °C, respectively, under 200 µmol m^−2^ s^−1^ light and recovered at normal temperature for 5 d. Controls represent the untreated-plants always kept in normal growth condition. (**a**) Phenotypes of plants after low temperature treatment. Photographs were taken after recovery for 5 d. (**b**) Effect of AMHA pretreatment on the cold injury index of tomato and strawberry plants treated with low temperature. (**c**) Effect of AMHA pretreatment on the shoot fresh weight of tomato and strawberry plants treated with low temperature. Values are the means ± SD of four independent biological replicates. Different small letters above error bars indicate significance at 0.05 level.

**Figure 7 ijms-23-05715-f007:**
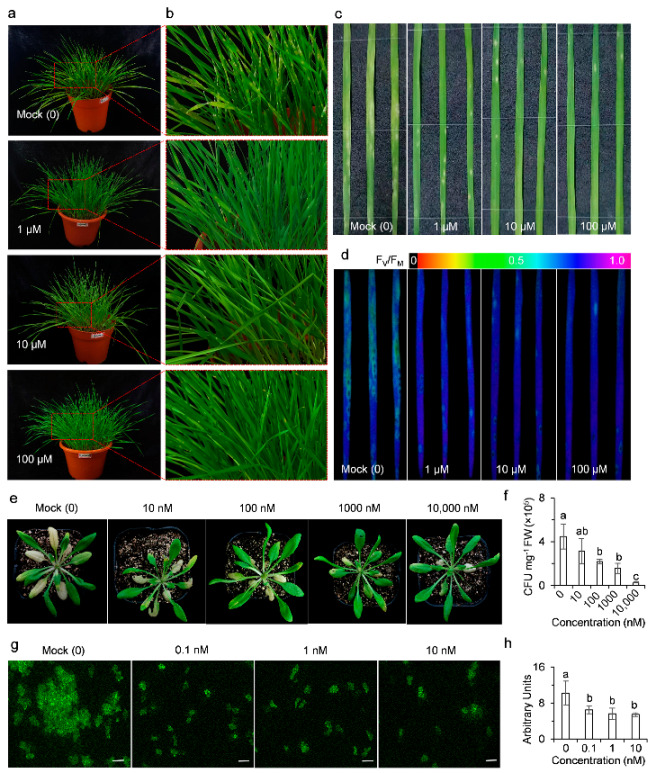
Induction of plant defense against pathogen attack by AMHA. Images of seedlings correspond to representative individuals from at least three independent biological replicates with similar results. (**a**–**d**) Pretreatment of AMHA confers resistance to powdery mildew in wheat. After pretreatment with water containing 0.02% Tween-20 (mock) or different concentrations of AMHA (1, 10 and 100 μM) containing 0.02% Tween-20, wheat seedlings were inoculated with conidia of *Blumeria graminis* f. sp. *tritici* (*Bgt*). Disease symptoms of wheat plants (**a**) and the indicated red pane parts (**b**) after inoculation for 8 d. Macroscopic infection phenotypes (**c**) and chlorophyll fluorescence images of F_V_/F_M_ (**d**) of representative leaves 8 d after inoculation of whole wheat plants with *Bgt*. Fluorescence images are color-coded based on the scale shown above the images. (**e**,**f**) Pretreatment of AMHA confers resistance to bacterial *Pst* DC3000 in *Arabidopsis*. After pretreatment with water containing 0.02% Tween-20 (mock) or different concentrations of AMHA (10, 100, 1000 and 10,000 nM) containing 0.02% Tween-20, *Arabidopsis* plants were inoculated with *Pst* DC3000. Disease symptoms developed by *Arabidopsis* plants 7 d after inoculation with *Pst* DC3000 (**e**). Concentration of bacteria in fresh leaves after 4 d inoculation of whole plants with *Pst* DC3000 (**f**). (**g**,**h**) Pretreatment of AMHA confers resistance of tobacco to TSWV. After pretreatment with water containing 0.02% Tween-20, tobacco leaves were infiltrated with *Agrobacterium*-mediated TSWV carrying SR_(+)eGFP_ + M_(−)opt_ + L_(+)opt_ constructs in vivo. eGFP fluorescence expression in tobacco leaves was visualized by a fluorescence microscope at 7 d (**g**) and quantified (**h**) to determine the amount of TSWV in the infected leaves. Scale bar = 200 μm. Results in F and H represent mean values ± SD of three independent biological replicates. Different small letters above error bars indicate significance at 0.05 level.

## Data Availability

The data presented in this study are available in the article and the Appendix A here.

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
