# Peer review of "Natural 2-Amino-3-Methylhexanoic Acid as Plant Elicitor Inducing Resistance against Temperature Stress and Pathogen Attack"

_ijms, 2022, doi:10.3390/ijms23105715_

Round 1

Reviewer 1 Report

The article by Wang et al. report the isolation of an α-amino acid, 2-Amino-3-methylhexanoic acid (AMHA),its chamical characterization and then its activity in plants, by increasing plant resistance to both abiotic and biotic stress.

The article is well constructed and easy to read. The topic is of high interest. I suggest the ms to be accepted after minor revisions. Please see my comments below.

  1. 52: Please check the difference between plant elicitor and biostimulant.
  2. 65: Is this ref necessary? I don’t think so.
  3. 103: please add “..chemical STRUCTURE…”

I have not the expertise to comment on the NRM and IR analyses, but in LCMS, a good and simple test is also running spiked samples (also for more accurate quantification).

Fig 3 and 4: What is the difference between 3c and 4a? How to explain the differences in RT? First, between strains (those are small, but regarding the short RT, they are surely meaningful…), and then between 3c and 4a? Could the authors comment?

Plant tests:

Please add precisions about the replicates:  what does “4 biological replicates” mean: is it 4 individual plants, or 4 pots (with several plants in each)? Please add this in the M&M: do the authors means: 3 tomato seeds PER POT (l. 501-503)? The AHMA concentrations used should also be listed in the M&M (§3.7).

  • 3.8-10: Why did the authors used different concentrations for each type of test?

I have a concern about the time between the last treatment and the inoculation (24h). The authors tested for the direct effect of the AMHA solution on growth of fungi and bacteria, but still, 24h is not enough for the plant to develop any defense priming. Washing the plants before inoculation would have been good. So the duration of the protection provided by AMHA appears as a key question for field application. This point really needs to be commented. As the authors mentioned, there is a lot of research questions open. 

Author Response

The article by Wang et al. report the isolation of an α-amino acid, 2-Amino-3-methylhexanoic acid (AMHA), its chamical characterization and then its activity in plants, by increasing plant resistance to both abiotic and biotic stress.

The article is well constructed and easy to read. The topic is of high interest. I suggest the ms to be accepted after minor revisions. Please see my comments below.

 RE: Thanks for your hard work and kind comments.

  1. 52: Please check the difference between plant elicitor and biostimulant.

RE: Thanks for your kind comments. We have modified this part (See Line 59).

  1. 65: Is this ref necessary? I don’t think so.

RE: We think this reference is necessary. Because it is the example of seaweed extract to improve plant tolerance to drought, salinity and heat stress conditions.

  1. 103: please add “..chemical STRUCTURE…”

RE: Thanks for your kind advice. We have modified this part (See Line 108).

I have not the expertise to comment on the NMR and IR analyses, but in LCMS, a good and simple test is also running spiked samples (also for more accurate quantification).

RE: Thanks. NMR and IR are the most effective and reliable technique to identify the structure of new chemicals.

Fig 3 and 4: What is the difference between 3c and 4a? How to explain the differences in RT? First, between strains (those are small, but regarding the short RT, they are surely meaningful…), and then between 3c and 4a? Could the authors comment?

REThanks for your hard work and kind comments.

(1) Figure 3c exhibited the data of the extracts from 7-day-old mycelia of four fungi cultured in Czapek solid medium, while the data in Figure 4a was about the extracts of four fungi cultivated for 7 d in Czapek liquid medium.

(2) There were perhaps three main reasons to explain the phenomenon of the differences in retention time (RT). Firstly, the range of RT was usually fluctuated at the range of ± 5%. It was often affected by the condition of LCMS instrument we used. In our serious of pre-experiments, the phenomenon of the RT of standard AMHA fluctuated also appeared sometimes. Secondly, although all the fungi we measured can produce AMHA, the variety of other compounds of extracts from different fungi may vary considerably. This might also affect the RT of AMHA. Thirdly, the different cultural conditions (solid or liquid medium) may also affect the compounds produced for the same fungi, which was similar to the second reason. In conclusion, the RT of AMHA in different fungi range from 1.58 to 1.74 min was normal in research.

Plant tests:

Please add precisions about the replicates:  what does “4 biological replicates” mean: is it 4 individual plants, or 4 pots (with several plants in each)? Please add this in the M&M: do the authors means: 3 tomato seeds PER POT (l. 501-503)? The AHMA concentrations used should also be listed in the M&M (§3.7).

RE: Thanks. We revised it according to your suggestion.

  • “Four or three biological replicates” mean that four or three pots with three or more plants were used (see the Line 538 to Line 539 and the Line 556 to Line 557, the Line 575 to Line 576 and the Line 590 to Line 591)  
  • Thirty wheat seeds, 0.2 g of bentgrass seeds or three tomato seeds per pot were planted (see the Line 508).
  • The AHMA concentration is 10, 100, 1000 and 10000 nM (see the Line 518).

3.8-10: Why did the authors used different concentrations for each type of test?

RE: Thanks very much. At the beginning of this research, we have done a series of pre-experiments. In order to find appropriate concentrations of AMHA to exhibit activity against different stress for plants. We set 100 nM as initial concentration of AMHA. It was indicated that AMHA could improve the wheat resistance to fungi powdery mildew and Arabidopsis resistance to bacterial infection by increasing treatment concentrations. On the contrary, the activity of AMHA for tobacco resistance to virus infection was declined at higher concentrations. So we used high concentrations of AMHA for eliciting wheat resistance against powdery mildew and Arabidopsis resistance to bacterial infection, and low concentrations of AMHA for eliciting tobacco resistance against virus in this research.

I have a concern about the time between the last treatment and the inoculation (24 h). The authors tested for the direct effect of the AMHA solution on growth of fungi and bacteria, but still, 24h is not enough for the plant to develop any defense priming. Washing the plants before inoculation would have been good. So the duration of the protection provided by AMHA appears as a key question for field application. This point really needs to be commented. As the authors mentioned, there is a lot of research questions open. 

RE: Thanks for your hard work and kind comments. In this study, seedlings were sprayed with AMHA twice or three times with a 24-h interval before stress treatment. This means it is 48 or 72 h after the first time spraying with AMHA when pathogens were inoculated on seedlings. Generally, the plants would appear obvious immunorection after 24 to 48 h of pretreatment in the greenhouse. To strengthen the immunoreaction of seedlings, here we added 1 or 2 times AMHA spraying with a 24-h interval. For field trial, AMHA was used to protect plant against stresses at ahead of 5 to 10 day of stress coming. This work will be exihibited in the next paper soon.

Reviewer 2 Report

Very interesting research that brings valuable information in enhancing plant resistance to biotic and abiotic stresses.

Research well planned and executed.
Results properly discussed.
Minor corrections should be made in the manuscript: 
1. Line 72: please include the publication number from the bibliography instead of the year of publication.
2. Lines 81-83: please further elaborate (explain) the purpose of the study.
3. Lines 84-90: this paragraph is appropriate for the conclusions and should not be in this part of the manuscript. 
4. More conclusions from the research should be presented.

Author Response

Very interesting research that brings valuable information in enhancing plant resistance to biotic and abiotic stresses.

Research well planned and executed.

Results properly discussed.

RE: Thanks for your hard work and kind comments.

Minor corrections should be made in the manuscript: 

  1. Line 72: please include the publication number from the bibliography instead of the year of publication.

RE: Thanks. We revised it according to your suggestion. Please see the Line 70.

  1. Lines 81-83: please further elaborate (explain) the purpose of the study.

RE: Thanks. We improved this part in the new version. Please see the Line 83 to Line 85.

  1. Lines 84-90: this paragraph is appropriate for the conclusions and should not be in this part of the manuscript. 

RE: Thanks for your good comments. We revised this part in the new version. Please see the Line 86 to Line 94.

  1. More conclusions from the research should be presented.

RE: Thanks. We added the new conclusion in the new version. Please see the Line 596 to Line 598.